# ^18^F-Fluoroethyl-L Tyrosine Positron Emission Tomography Radiomics in the Differentiation of Treatment-Related Changes from Disease Progression in Patients with Glioblastoma

**DOI:** 10.3390/cancers16010195

**Published:** 2023-12-30

**Authors:** Begoña Manzarbeitia-Arroba, Marina Hodolic, Robert Pichler, Olga Osipova, Ángel Maria Soriano-Castrejón, Ana María García-Vicente

**Affiliations:** 1Nuclear Medicine Department, University Hospital of Toledo, 45007 Toledo, Spain; begona.manzarbeitia@gmail.com (B.M.-A.); a_sor_cas@yahoo.es (Á.M.S.-C.); 2Nuclear Medicine Department, Faculty of Medicine and Dentistry, Palacky University, 779 00 Olomouc, Czech Republic; marina.hodolic@gmail.com; 3Institute of Nuclear Medicine Kepler University Hospital—Neuromed Campus, 4020 Linz, Austria; robert.pichler@kepleruniklinikum.at (R.P.); olga.osipova@kepleruniklinikum.at (O.O.)

**Keywords:** ^18^F-fluoroethyl-tyrosine, positron emission tomography, radiomics, glioma tumor progression, treatment-related changes

## Abstract

**Simple Summary:**

^18^F-Fluoroethyl-L tyrosine radiomics are useful in the differentiation of true progression from treatment-related changes in patients with glioblastoma, offering relevant complementary information with respect to the reference standard magnetic resonance imaging.

**Abstract:**

The follow-up of glioma patients after therapeutic intervention remains a challenging topic, as therapy-related changes can emulate true progression in contrast-enhanced magnetic resonance imaging. ^18^F-fluoroethyl-tyrosine (^18^F-FET) is a radiopharmaceutical that accumulates in glioma cells due to an increased expression of L-amino acid transporters and, contrary to gadolinium, does not depend on blood–brain barrier disruption to reach tumoral cells. It has demonstrated a high diagnostic value in the differentiation of tumoral viability and pseudoprogression or any other therapy-related changes, especially when combining traditional visual analysis with modern radiomics. In this review, we aim to cover the potential role of 18F-FET positron emission tomography in everyday clinical practice when applied to the follow-up of patients after the first therapeutical intervention, early response evaluation, and the differential diagnosis between therapy-related changes and progression.

## 1. Introduction

Despite new molecular concepts introduced in the last decade, the diagnostic approach of patients with glioblastoma multiforme (GBM) is still conventional, mainly through the maintenance of contrast-enhanced magnetic resonance imaging (MRI) as the reference imaging technique for diagnosis, biopsy guidance, and treatment planning as well as treatment monitoring. So, despite the developments of new sequences for diffusion-weighted imaging (that assess the movement of water molecules and provide information on the cellularity of the tumor) and perfusion imaging (that provides information on tissue perfusion and permeability), no significant modifications in prognosis have been detected in the last decades. In addition, the inability of current treatments to achieve disease control explains why about 80% of GBM relapses occur within a 2 cm margin from the enhancing tumor location [1].

Positron emission tomography (PET) with labeled amino acids has been recommended to guide GBM resection and to delineate GBM extent by the Response Assessment in Neuro-Oncology (RANO) Working Group [2,3], based on a higher accuracy compared to other radiotracers and MRI. However, PET usually serves as a second-line diagnostic modality in neuro-oncology, performed only on the recommendation of a multidisciplinary tumor board in a minority of cases, mainly during the disease course, when patients present uncertain MRI features or an equivocal clinical course after or during treatment.

^18^F-fluoroethyl-L tyrosine (^18^F-FET) is an amino acid PET radiotracer that accumulates in glioma cells due to an increased expression of L-amino acid transporters (LAT) 1 and 2 as well as an increased tumor perfusion [4]. Moreover, a disruption of the blood–brain barrier (BBB) can also lead to a passive influx of the radiotracer to the tumoral tissue, although this is not a prerequisite for the intratumoral accumulation of ^18^F-FET [1].

In Europe, commercialized ^18^F-FET (IASOglio^®^), only authorized in France and Poland, includes the following indications of use in patients with glioma: characterization of brain lesions suggestive of glioma and selecting the best biopsy site in them, noninvasive grading of glioma, pretherapeutic delineation of viable glioma tissue, and after treatment, for the detection of viable tumor tissue in case of the suspicion of a persistent or recurrent glioma [5].

However, the differential diagnosis of pseudoprogression (PsP) from true progression (TP) is a big challenge in clinical practice for patients with a prior history of radical treatment and sometimes can only be made retrospectively based on an MRI follow-up. ^18^F-FET has demonstrated high efficacy in the diagnosis of PsP when compared to other PET radiopharmaceuticals and even to MRI [2,4].

Radiomics is the extraction of quantitative characteristics from medical images using advanced data-characterization algorithms. Radiomic features derived from PET have been described as an effective tool in the prediction of molecular tumor characteristics and patient outcome in several tumors [6]. Textural variables (first-, second-, and third-order parameters) that define the relations between voxel uptake and other measures of a more global spatial heterogeneity, such as the coefficient of variation that analyzes the spatial dispersion of gray intensity levels in voxels, have been defined. In addition, the SUV mean offers more integrated information of the radiotracer uptake in tumor voxels than the maximum standardized uptake value (SUVmax), and the SUVmean/SUVmax ratio can be described as an additional variable showing a direct relationship with homogeneity. On the other hand, tumor shape features, such as sphericity, seem to refer to the infiltrative tumor capacity of the tumor, so a less spherical lesion can be associated to a more aggressive molecular pattern. Moreover, other variables that inform about the molecular or metabolic tumor burden can be addressed by PET as the metabolic (or biological) tumor volume (MTV), that is, the volume of interest (VOI) after segmentation and total lesion activity (TLA) defined as the product of SUVmean by MTV.

In addition, radiomics extracted from 18F-FET PET have been found to add valuable diagnostic information in the prediction of the isocitrate dehydrogenase (IDH) enzyme genotype, the differential diagnosis between PsP and TP, and the prognosis prediction in newly diagnosed gliomas [7].

In the following sections, methodological, clinical aspects and controversies are described, referring to the most relevant and recent literature. Some illustrative cases are presented in Figure 1, Figure 2 and Figure 3.

## 2. Methodological Aspects

### 2.1. Imaging Acquisition and Analysis

The current protocol for ^18^F-FET PET/CT includes both static and dynamic acquisition [2]. A static scan is usually obtained 20 min after 18F-FET injection. However, a study by Verburg et al. [8] suggests performing ^18^F-FET PET 60–90 min after tracer injection in order to obtain a higher diagnostic accuracy. Dynamic acquisition is recommended and should be started right after the radiotracer injection, using 30–60 s frames within the first 10 min and 5 min frames within the next 40–50 min. The evolution of ^18^F-FET uptake as a VOI in a function of time can be presented as a time–activity curve (TAC).

A static study can be visually analyzed, although the obtention of the semiquantitative data of ^18^F-FET as the SUVmax or SUVmean is very informative. Based on a faint physiologic uptake in healthy brain, an increased radiotracer uptake can be described also using the tumor-to-background ratio (TBR). For that reason, a VOI delineating the tumor, and any area of healthy brain, preferable on the contralateral hemisphere, is selected. Afterwards, the SUVmax or SUVmean of tumor tissue is divided by the SUVmean of healthy brain.

The accurate tumor delineation of ^18^F-FET PET is crucial for interpretation. The usual threshold for the definition of the biological tumor volume (BTV) is 1.6 in static studies [9,10]. However, there is not a validated threshold and either standardization regarding the tumor and background VOIs. The largest reported FET-PET-credentialing study detected considerable variability in BTV delineation and image interpretation, even using the fixed value of 1.6. The discordances were explained by the manual adjustment of the segmented volume to remove any obvious non-tumor structures. However, despite these discrepancies, TBRmax and TBRmean were robust variables [11].

Several reports have shown that some radiomic features are sensitive to variations in several factors, including image acquisition, image reconstruction, and tumor segmentation, as well as test–retest imaging [11,12]. Zounek et al. [13] quantified the sensitivity of radiomic features derived from ^18^F-FET PET images of glioma patients with respect to variations in image reconstruction settings and tumor segmentation methods. The overall results showed that PET radiomic features, especially those shape-derived, were highly sensitive to the choice of image segmentation methods. On the other hand, Gutsche et al. [14] defined that first-order features extracted from the image histogram showed the highest repeatability. They also found a correlation between tumor volume and feature repeatability (for tumor volumes larger than 4 mL, more than 50% of features showed high repeatability), and a comparable repeatability was found between IDH-wild-type and IDH-mutant gliomas (repeatable features, 63% vs. 52%, respectively).

### 2.2. ^18^F-FET PET Interpretation

^18^F-FET scans should be fused with a CT or recent T2-weighted and T2/FLAIR-weighted MRI for a proper analysis of the lesions. The interpretation must be both visual and semiquantitative (such as TAC or TBR), as the sum of both analyses increases diagnostic accuracy and ensures intra-and inter-individual comparability.

The curve pattern of the dynamic acquisition is important, especially for lesion characterization. Three patterns have been described: ^18^F-FET uptake without an identifiable peak uptake (pattern I); ^18^F-FET uptake peaking at a midpoint (>20–40 min) followed by a plateau or a small descent (pattern II); and ^18^F-FET uptake peaking early (≤20 min) followed by a constant descent (pattern III) [15]. Pattern III is characteristic for high-grade gliomas (HGGs) or areas of malignant transformation in low-grade gliomas (LGGs) [16].

However, ^18^F-FET uptake is not always specific for neoplastic tissue, and pitfalls have been reported for brain abscesses, demyelinating processes, in the proximity of cerebral ischemic lesions, hematomas, and even in areas of reactive astrogliosis after high-dose brachytherapy [17,18]. In dexamethasone treatment, a possible increase in ^18^F-FET uptake by normal brain tissue has been described. In addition, a temporary increase in ^18^F-FET gyral uptake in the peri-ictal period of epilepsy may mimic a focal lesion [19].

## 3. Differentiation of Tumor Progression and Therapy-Related Changes

### 3.1. Pseudoprogression

In clinical practice, PsP is of considerable importance, representing approximately one-third of patients with GBM, usually within the first 12 weeks after radiochemotherapy with temozolomide (RCT-TMZ) [20,21]. PsP is a consequence of treatment-related local tissue reactions which comprise inflammation, oedema, and an increased permeability of the BBB, thereby resembling TP. PsP is associated with a better outcome, so if it is suspected, treatment should not be stopped, as it improves survival in patients with PsP [22,23,24]. Thus, a correct diagnosis is essential to avoid terminating an effective therapy. In patients with PsP, ^18^F-FET uptake is significantly lower than in patients with TP [25].

Magnetic resonance imaging using the RANO criteria, which relies heavily on findings such as contrast enhancement, T2-weighted and FLAIR changes, and further characterizes measurable versus non-measurable disease, is currently the reference standard in the diagnosis and follow-up of glioma patients. In the post-therapy setting, PsP appears as an enlarged area of contrast enhancement on MRI, and it cannot be effectively identified on a single MRI [24], being necessary sequential studies to demonstrate its subsequent improvement or stabilization without treatment [15]. Advancements in MRI, particularly perfusion-weighted imaging (PWI) using regional cerebral blood volume (rCBV) or diffusion-weighted imaging (DWI), can contribute to the differentiation of TP and therapy-related changes. However, treatment-related inflammation increases rCBV, and radiation necrosis may show as diffusion restriction due to oedema and leukocyte infiltrates in the transition zones [26,27].

In a Bayesian network meta-analysis including different PET radiotracers and MRI for recurrent glioma, ^18^F-FET showed the highest sensitivity, specificity, positive predictive value, and accuracy [4]. However, when a simultaneous imaging of ^18^F-FET PET and MRI obtained by a hybrid PET–MRI system comes into play, no significant differences seem to exist, probably explained by an increase in the interpretation confidence of both joined techniques [28,29].

The advantage of metabolic imaging using ^18^F-FET is explained because in tissue affected by post-therapeutic changes, LAT expression is normal or even downregulated (contrary to glioma tissue, which presents an increased number of LAT). However, an increased uptake of 18F-FET in PsP tissue can also appear, as a consequence of the passive influx of ^18^F-FET through a disrupted BBB, although with a lower intensity than TP [30].

In a recent meta-analysis, the pooled sensitivity and specificity of ^18^F-FET using a TBRmax of 1.9–2.3, for the differentiation of glioma recurrence from treatment-related changes, were 91% (95% CI, 74–97%) and 84% (95% CI, 69–93%), respectively [31]. Furthermore, an optimal cut-off value of 2.3 (accuracy 96%, area under curve of 0.94 ± 0.06; *p* < 0.001) has been described for identifying PsP, being significantly predictive of a longer overall survival (OS median 23 vs. 12 months; *p* = 0.046). Conversely, it seems advisable to assume late PsP when TBRmax is below 1.0. On the other hand, values between 1.0 and 2.3 should be interpreted with caution, as there is an overlap of final diagnoses [15].

In addition, dynamic ^18^F-FET PET may be helpful in the differentiation between TP and treatment-related changes or gliosis [32]. PsP, reactive gliosis, and benign tissues are associated with a TAC pattern of a steadily increasing ^18^F-FET uptake without an identifiable peak. In addition, a TAC pattern with an early or midpoint time to peak (TTP) uptake followed by a constant decline or plateau has been described as being highly specific of TP [10,25].

In current clinical practice, static ^18^F-FET PET parameters (TBRmean and TBRmax) seem to have a superior diagnostic performance in comparison to dynamic parameters in the detection of recurrence, although a combination of static and dynamic ^18^F-FET PET is the most valuable [33,34]. Additionally, other radiomics, informative of the heterogeneity of voxel radiotracer distribution, are associated with TP instead of PsP [5].

Regarding textural features on ^18^F-FET, small zone/low gray emphasis may be helpful in predicting the time to progression in patients with recurrent GBM undergoing re-irradiation [35].

IDH mutation status may also be taken into account when performing ^18^F-FET PET, as it has been described that the diagnostic accuracy when differentiating PsP and TP may depend on the mutation status, with higher accuracies observed in those with IDH-wild-type GBM [36,37].

### 3.2. Pseudoresponse

Antiangiogenic chemotherapeutics, such as bevacizumab (BEV), work by reducing the vascular permeability of immature blood vessels supplying the tumor and repairing the BBB. This results in a decrease in contrast enhancement in the peritumoral edema, regardless of the tumor’s sensitivity to the drug, which limits the diagnostic capability of MRI, hindering the differentiation between a good response to treatment from treatment-induced change [38].

^18^FET PET has demonstrated a better accuracy for disease monitoring than conventional MRI in BEV-treated patients with glioma [39,40]. Furthermore, adding ^18^F-FET PET to MRI can increase the rate of correct diagnoses by 41% [32]. However, the response criteria are not standardized in ^18^F-FET PET. Galldiks et al. [41] used the following criteria to identify responders in dynamic ^18^F-FET PET: (i) an increase in TTP from the baseline to the follow up PET scan greater than or equal to 10 min, (ii) a baseline TTP greater than 25 min, and (iii) a kinetic pattern of either an SUV peak at the end of the study or a TTP in the middle of the study followed by a plateau or slow descent. Other authors identified a decrease in ^18^F-FET PET/CT tumor volumes as a sign of BEV response [42].

However, with independence of the different criteria used, a response to treatment identified on ^18^F-FET PET seems to predict an increase in OS and progression free survival (PFS) of treated patients [41,42]. On the other hand, ^18^FET PET may detect tumor progression during antiangiogenic treatment earlier than MRI. Wirsching et al. [43] described that high ^18^FET-TBR of non-contrast-enhancing tumor portions during BEV therapy was associated with inferior OS on multivariate analysis (HR 5.97; CI, 1.16–30.8).

### 3.3. Early Response Evaluation

Previous studies have defined that MTV changes are predictive for the early identification of metabolic responders in patients undergoing adjuvant TMZ chemotherapy or lomustine-based chemotherapy in recurrent gliomas [14,44,45].

Suchorska et al. [46] proposed the following classification scheme for the evaluation of treatment response. In patients with non-contrast-enhancing glioma, a responsive disease was defined when a decrease in either BTV ≥ 25% and/or TBRmax ≥ 10% occurred; an increase in BTV ≥ 25% and/or TBRmax increase > 10% characterized a progressive disease, and minor changes ±25% for BTV and ±10% for TBRmax were regarded as a stable disease. Using the previous criteria, patients with a responsive disease had the longest time to treatment failure, while there was no significant difference between patients with a stable disease and progressive disease. On the contrary, a T2-volume-based assessment was not associated with outcome. Thus, in contrast to gadolinium-volume changes in MRI, changes in ^18^F-FET PET may be a valuable parameter to assess treatment response in GBM and predict survival [47].

### 3.4. Molecular Dependence

O6-methylguanine DNA methyltransferase (MGMT) gene promoter methylation and IDH mutation status allow for the stratification into biologically and prognostically distinct subgroups of glioma patients, PsP being more frequent [48]. Additionally, mutation status can be predicted by unifying dynamic and static features using TTP combined with the TBR max or TAC slope [49]. In patients with an MGMT methylated status, RCT-TMZ is effective in controlling GBM cells in the tumoral bed but not in controlling distant recurrence. In MGMT unmethylated patients, the rate of in-field recurrences is higher, which leaves room for a dose escalation with modern radiation techniques [50]. After the administration of TMZ concomitant with and adjuvant to RT in patients with GBM, the relapse pattern determined by ^18^F-FET PET/CT has been associated with the MGMT methylation status, with a higher PFS and ex-field recurrence rate in MGMT methylated patients, which might be a sign of better local control [51].

In addition, radiomic textural features seem to outperform the traditional ^18^F-FET parameters, such as TBRmax or TBRpeak, in the prediction of mutation status and for the differentiation of treatment-related changes from TP in GBM with enough confidence [52,53].

## 4. Comparison with Other PET Radiotracers

Many PET radiopharmaceuticals have been studied for brain tumor management. However, the consideration of a wide spectrum of primary brain tumors as a single pathological entity may prevent us from discovering the full potential of each radiopharmaceutical at our disposal. Taking these differences into account may allow us to reach more solid conclusions.

### 4.1. Fluorodeoxyglucose (FDG)

^18^F-FDG has a high physiological uptake in brain tissue and a relatively low uptake in some specific histological subtypes, such a lower-grade gliomas, which brings about a low lesion-to-background contrast. Delayed images can increase the target-to-background contrast, increasing the diagnostic accuracy. However, ^18^F-FET outperforms ^18^F-FDG in the differential diagnosis between radiation necrosis and recurrent disease in irradiated diffuse gliomas, with a sensitivity of 82–91% and 70–84%, respectively, and a specificity of 78–95% for FET and 70–88% for ^18^F-FDG [54].

### 4.2. Choline Analogues

The in vitro kinetic uptake of ^18^F-fluorocholine (^18^F-FCH) and ^18^F-FET is quite similar, only demonstrating a difference in uptake velocity: ^18^F-FET shows a more rapid initial uptake up to 40 min, and ^18^F-FCH shows a more progressive, continuous rise reaching a maximum activity plateau after 90 min [55]. However, the cellular transport mechanism of choline analogs differs with respect to amino acid tracers. First, ^18^F-FET uptake is mediated by LAT, whereas ^18^F-FCH uptake correlates with choline transporter-like 1 expression [8]. Second, ^18^F-FCH metabolism is very fast, with the parent fraction of the tracer decreasing in 15 min to 27% [8], compared to 87% in 120 min for ^18^F-FET, resulting in a better tracer availability [8,34]. Third, capillary density has also been described as influential in ^18^F-FCH uptake but not in ^18^F-FET uptake [56], and finally, the dependency of ^18^F-FET uptake on the breakdown of the BBB is less than that of ^18^F-FCH, with a high ^18^F-FET uptake also seen in tumor regions outside the area of contrast enhancement [57].

Moreover, the presence of reparative changes after therapy acts in a different manner in both radiotracers, with a higher affinity of ^18^F-FCH to inflammatory cells and an up-regulation of choline-kinase and/or choline transporters and a lower upregulation of LAT caused by radiation [56,58].

### 4.3. Other Amino Acid Tracers

Different tracer kinetics in malignant and benign tissues appear to be a special property of ^18^F-FET and have not been observed for other amino acid tracers such as 11C-methyl-L-methionine (^11^C-MET) or 3,4-dihydroxy-6-[18F]-fluoro-l-phenylalanine (^18^F-FDOPA) [59,60].

^11^C-MET and ^18^F-FDOPA provide comparable diagnostic information compared to ^18^F-FET for the differentiation between residual or recurrent tumor and treatment-related changes/PsP, as well as the delimitation of gliomas, although ^18^F-FET shows higher SUVs and a TBR mean for HGG than ^18^F-DOPA [61,62].

### 4.4. Prostate-Specific Membrane Antigen Ligands

Preliminary clinical results showed significantly high values of the in vivo uptake of prostate-specific membrane antigen (PSMA) ligands into HGG [63]. In a prospective study using ^68^Ga-PSMA-617 in a small sample of patients with recurrent glioma, ^68^Ga-PSMA-617 accumulated in large parts of the tumor that extended beyond the ^18^F-FET-avid margins, suggesting that PSMA ligands target a complementary biological process and might be a useful diagnostic marker to delineate parts of the recurrent tumor that are neoangiogenic but not extremely metabolically active yet. In addition, ^68^Ga-PSMA-617 had a higher TBR than ^18^F-FET, suggesting a better tumor specificity of the former [64].

## 5. Future Directions and Conclusions

Although MRI remains the standard of care, given the lack of alternatives available in current clinical practice, ^18^F-FET provides complementary information regarding the treatment response after chemoradiation, in terms of the prognostication of recurrence and patient survival. However, some pending issues deserve consideration.

The supplemental value of ^18^F-FET with respect to standard MRI must be addressed in prospective studies. So, we expect that the ongoing ^18^FET PET in glioblastoma (FIG) study [65], designed to determine the accuracy and management impact of ^18^F-FET PET in several clinical settings, reveals robust results.

The standardization and criteria harmonization in the imaging evaluation and interpretation of ^18^FET PET is mandatory, so credentialing studies are necessary to increase the expertise level across study sites.

Regarding radiomic features, before their implementation in clinical practice, it is essential to ensure their reproducibility and robustness. Thus, to properly translate radiomic models into clinical routine, they should be validated on large datasets that preferably include data from multiple centers.

The diagnostic impact of ^18^F-FET PET attending to the different molecular tumor profiles should be addressed based on their expected interdependency.

The clinical value of MTV must be demonstrated, in order to include volumetric assessment in consensus guidelines and recommendations.

Summarizing, regarding clinical decision making, ^18^F-FET radiomics may offer relevant information for patients with glioma, being useful in the differentiation of TP from therapy-related changes, overcoming the limitations of conventional MRI.

## Figures and Tables

**Figure 1 cancers-16-00195-f001:**
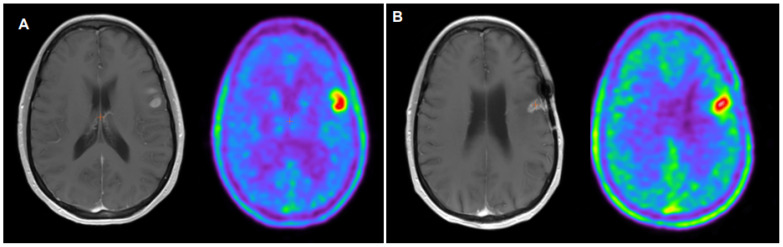
45-year-old female diagnosed with an IDH1-2 wild-type GBM (WHO grade 4, TERT mutation, EGFR amplification, promotor MGMT unmethylated). Preoperative MRI and 18F-FET PET (**A**) showed an increased 18F-FET uptake in a contrast-enhanced left frontal lesion on MRI with a TBR max and mean of 3.2 and a 2.4, respectively. A complete surgical resection was performed followed by Stupp protocol and adjuvant temozolomide. Thirteen months later, 18F-FET PET (**B**) was performed to rule out tumor relapse after a previous suspicious MRI, showing an increased radiotracer uptake in the left frontal lobe, close to the postsurgical changes, with a TBRmax of 1.9 and a TBRmean of 1.7, consistent with recurrent disease. The patient denied any specific tumor therapy, and palliative corticoid therapy was initiated.

**Figure 2 cancers-16-00195-f002:**
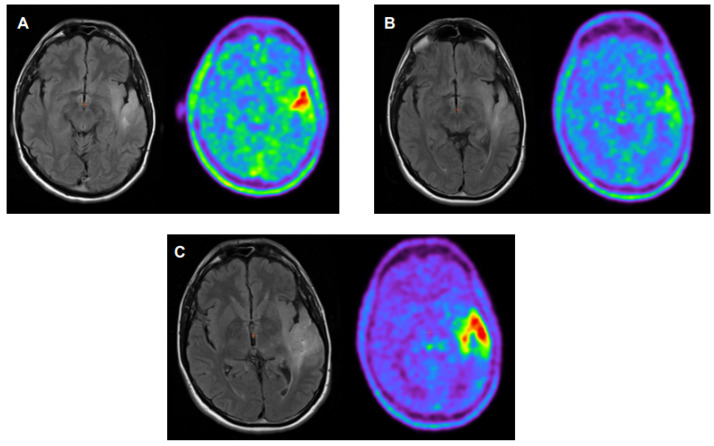
45-year-old male diagnosed with a grade II glioma by stereotactic biopsy. Preoperative MRI (FLAIR) and 18F-FET PET (**A**) showed an increased radiotracer uptake in a non-contrast-enhanced lesion on the left temporal lobe, with a TBRmax and mean of 2.4 and 2.0, respectively. Therapy with temozolomide and additional radiotherapy was administered. (**B**) Control MRI showed a stable disease with signs of a partial response to 18F-FET PET (TBRmax and TBRmean of 1.0 and 0.9, respectively). Following MRI was suspicious of progression and a new 18F-FET PET was performed (**C**), showing an increased 18F-FET uptake in the left temporal lobe, more intense and broader than in any previous study (TBR max and TBR mean of 3.8 and 3.1, respectively) and consistent with disease progression. A second-line therapy with temozolomide was scheduled; however, a progressive disease was detected on the last follow-up MRI, considered a probable transformation to a high-grade glioma based on disease evolution.

**Figure 3 cancers-16-00195-f003:**
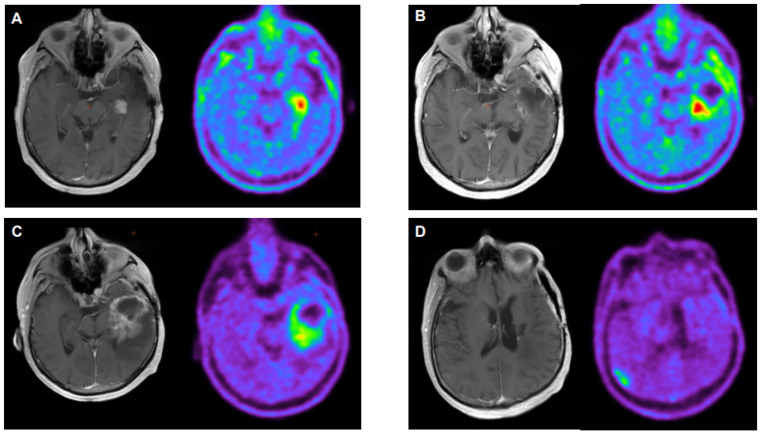
63-year-old male diagnosed with a left temporal glioblastoma (WHO grade IV, IDH wild type, no EGFR amplification, MGMT promoter unmethylated). The patient was treated with a tumor resection and posterior chemoradiotherapy with adjuvant temozolomide. Three years after the initial diagnosis, and with a suspicion-of-recurrence MRI, the patient underwent 18F-FET PET, showing an increased radiotracer uptake in the left temporal lobe (TBRmax and TBRmean of 2.5 and 2.3, respectively), suspicious of neoplastic tissue (**A**). Surgery was performed, confirming the relapse. A year later, an MRI showed signs of tumor regrowth but partially undistinguishable from radiation necrosis. 18F-FET PET (**B**) showed uptake in the left temporal lobe, adjacent to the surgical cavity (TBRmax and TBRmean of 2.6 and 2.3, respectively), consistent with disease progression. The patient underwent additional radiotherapy. Six months later, tumor regrowth appeared in the follow-up MRI with doubts of pseudoprogression, whereas 18F-FET PET showed signs of progressive disease with an increased ring-like shape uptake (TBRmax and TBRmean 3.4 of 2.8, respectively) in the left temporal lobe (**C**) and a new focal uptake (TBRmax and TBRmean of 2.5 and 2.1, respectively) on the contralateral parietal lobe (**D**), not observed in a contrast-enhanced MRI. A new therapeutic line with bevacizumab was initiated.

## Data Availability

Data sharing not applicable.

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
