# Peer review of "18F-Fluoroethyl-L Tyrosine Positron Emission Tomography Radiomics in the Differentiation of Treatment-Related Changes from Disease Progression in Patients with Glioblastoma"

_cancers, 2023, doi:10.3390/cancers16010195_

Round 1

Reviewer 1 Report

Comments and Suggestions for Authors

This is a simple review of FET PET for the follow-up imaging of patients with glioblastoma. 

Three cases were presented, however, the clinical courses are progressing. Imaging findings should be confirmed by pathological examination or long-term clinical courses.

The title should not include any abbreviation. After the definition of any abbreviation, the author should use the abbreviations.

Author Response

Authors, thank a lot the extensive review and the comments of all the reviewers.

A deep review has been performed taking into account all the comments, including grammar.  All the changes are coloured in blue into the manuscript.  Derived of the extensive review, a lot of references have been changed and added. They are in blue color too.

Text has been expanded to achieve the word number requirements.

The author’s comments are underlined in this document

Reviewer 1

This is a simple review of FET PET for the follow-up imaging of patients with glioblastoma.

Three cases were presented, however, the clinical courses are progressing. Imaging findings should be confirmed by pathological examination or long-term clinical courses. Yes, we asked for collaboration to other colleagues regarding clinical cases. Histopathology is scarcely used in the follow-up setting because it is difficult to performed and with a lot of risks. So, in current clinical practice, diagnosis of active disease is established when two diagnostic imaging techniques are positive or suspicious of. On the other hand, if one is doubtful, for example the MRI and PET is negative, follow-up is decided, without introduction of any additional therapy. Patient of figure 1, denied specific treatment (a brief comment has been added). Patient 2 was confirmed by imaging and patient 3 by surgery and imaging. All the explanations are included in figure legends.

The title should not include any abbreviation. After the definition of any abbreviation, the author should use the abbreviations. Done

Reviewer 2 Report

Comments and Suggestions for Authors

Ana Maria Garcia-Vicente and the coauthors provide an article with a comprehensive review of 18F-FET PET in the differentiation of treatment-related changes from disease progression in patients with glioblastoma. They have covered different methodological and clinical aspects of the topic, including imaging acquisition, interpretation, and comparison with other PET radiotracers. The manuscript is well written and provides an insightful analysis of the current literature. It can be accepted after addressing the following issues:

1. Can the authors provide more informaiotn on reproducibility and reliability of the radiomics features extracted from 18F-FET PET?

2. Can the authors include more discussion on advantages, limitations, and potential applications of the PET radiotracers for comparison?

3. Please check the author list. No corresponding author now in the manuscript.

4. Definition of PET should be added on Line 24. There are some clerical errors. Please do proof reading.

Author Response

Authors, thank a lot the extensive review and the comments of all the reviewers.

A deep review has been performed taking into account all the comments, including grammar.  All the changes are coloured in blue into the manuscript.  Derived of the extensive review, a lot of references have been changed and added. They are in blue color too.

Text has been expanded to achieve the word number requirements.

The author’s comments are underlined in this document.

Reviewer 2

Ana Maria Garcia-Vicente and the coauthors provide an article with a comprehensive review of 18F-FET PET in the differentiation of treatment-related changes from disease progression in patients with glioblastoma. They have covered different methodological and clinical aspects of the topic, including imaging acquisition, interpretation, and comparison with other PET radiotracers. The manuscript is well written and provides an insightful analysis of the current literature. It can be accepted after addressing the following issues:

  1. Can the authors provide more information on reproducibility and reliability of the radiomics features extracted from 18F-FET PET? Yes, a lot of paragraphs have been added to this respect.
  2. Can the authors include more discussion on advantages, limitations, and potential applications of the PET radiotracers for comparison? Other radiotracers have been included.
  3. Please check the author list. No corresponding author now in the manuscript. It has been Included.
  4. Definition of PET should be added on Line 24. There are some clerical errors. Please do proof reading.

Reviewer 3 Report

Comments and Suggestions for Authors

In this review, authors aim to cover the potential role of 18F-FET PET in everyday clinical practice when applied to the follow-up of patients after the first therapeutical intervention, early response evaluation and the differential diagnosis between therapy-related changes and progression.

This manuscript is actually just a constellation of already well-known, previously published information. Nothing critical or new can be found. 

According to an article entitled "Review articles: purpose, process, and structure" [1], the objective of a review article should be one or more of the following: (A) Resolve definitional ambiguities and outline the scope of the topic, (B)Provide an integrated, synthesized overview of the current state of knowledge, (C) Identify inconsistencies in prior results and potential explanations, (D)Evaluate existing methodological approaches and unique insights, (E) Develop conceptual frameworks to reconcile and extend past research, or (F) Describe research insights, existing gaps, and future research directions.

Unfortunately, none of these objectives is achieved by the manuscript herein.

References:

[1] Palmatier RW, Houston MB, Hulland J. Review articles: purpose, process, and structure. J. of the Acad. Mark. Sci. (2018) 46:1-5. DOI 10.1007/s11747-017-0563-4

Comments on the Quality of English Language

Moderate editing of English language required 

Author Response

Authors, thank a lot the extensive review and the comments of all the reviewers.

A deep review has been performed taking into account all the comments, including grammar.  All the changes are coloured in blue into the manuscript.  Derived of the extensive review, a lot of references have been changed and added. They are in blue color too.

Text has been expanded to achieve the word number requirements.

The author’s comments are underlined in this document.

Reviewer 3

In this review, authors aim to cover the potential role of 18F-FET PET in everyday clinical practice when applied to the follow-up of patients after the first therapeutical intervention, early response evaluation and the differential diagnosis between therapy-related changes and progression.

This manuscript is actually just a constellation of already well-known, previously published information. Nothing critical or new can be found. Ok, we have included new points of view of recent papers and a more critical overview.

According to an article entitled "Review articles: purpose, process, and structure" [1], the objective of a review article should be one or more of the following: (A) Resolve definitional ambiguities and outline the scope of the topic, (B)Provide an integrated, synthesized overview of the current state of knowledge, (C) Identify inconsistencies in prior results and potential explanations, (D)Evaluate existing methodological approaches and unique insights, (E) Develop conceptual frameworks to reconcile and extend past research, or (F) Describe research insights, existing gaps, and future research directions.

Unfortunately, none of these objectives is achieved by the manuscript herein. Ok, we respect your point of view, although several points that you referred were included. However, we have included and discussed more controversial issues and added some new points in the last part of manuscript (future directions).

References:

[1] Palmatier RW, Houston MB, Hulland J. Review articles: purpose, process, and structure. J. of the Acad. Mark. Sci. (2018) 46:1-5. DOI 10.1007/s11747-017-0563-4

Comments on the Quality of English Language

Moderate editing of English language required. The paper has been revised  and corrected .

Reviewer 4 Report

Comments and Suggestions for Authors

The authors reviewed well the role and significance of 18F-FET PET in the differentiation of true progression from treatment related changes in patients with glioblastoma. FET PET offering a relevant complementary information with respect to the reference standard magnetic resonance imaging. This review paper is well organized and easy to follow, which is acceptable to publish.

 However, one thing flaws the value of this paper is that there is a discrepancy between the title and content of the paper. Although radiomic is highlighted in the title, there is not much about FET radiomic in the actual content. In general, radiomics is a quantitative approach to medical imaging, extracts a large number of features from medical images using algorithms. Therefore, this method enhancing the existing data available to clinicians by means of advanced. The content of radiomic approach for FET PET covered in the paper is too weak, and the value of the paper will increase if this part is further strengthened.

Author Response

Authors, thank a lot the extensive review and the comments of all the reviewers.

A deep review has been performed taking into account all the comments, including grammar.  All the changes are coloured in blue into the manuscript.  Derived of the extensive review, a lot of references have been changed and added. They are in blue color too.

Text has been expanded to achieve the word number requirements.

The author’s comments are underlined in this document.

Reviewer 4

The authors reviewed well the role and significance of 18F-FET PET in the differentiation of true progression from treatment related changes in patients with glioblastoma. FET PET offering a relevant complementary information with respect to the reference standard magnetic resonance imaging. This review paper is well organized and easy to follow, which is acceptable to publish.

 However, one thing flaws the value of this paper is that there is a discrepancy between the title and content of the paper. Although radiomic is highlighted in the title, there is not much about FET radiomic in the actual content. In general, radiomics is a quantitative approach to medical imaging, extracts a large number of features from medical images using algorithms. Therefore, this method enhancing the existing data available to clinicians by means of advanced. The content of radiomic approach for FET PET covered in the paper is too weak, and the value of the paper will increase if this part is further strengthened. Yes, some paragraphs referring radiomics have been added.

Round 2

Reviewer 1 Report

Comments and Suggestions for Authors

I found significant improvement of this manuscript.